# Free and Added Sugar Consumption and Adherence to Guidelines: The UK National Diet and Nutrition Survey (2014/15–2015/16)

**DOI:** 10.3390/nu12020393

**Published:** 2020-02-01

**Authors:** Birdem Amoutzopoulos, Toni Steer, Caireen Roberts, David Collins, Polly Page

**Affiliations:** 1NIHR Biomedical Research Centre (BRC) Diet, Anthropometry and Physical Activity Group, MRC Epidemiology Unit, University of Cambridge, Cambridge CB2 1TN, UK; toni.steer@mrc-epid.cam.ac.uk (T.S.); caireen.roberts@mrc-epid.cam.ac.uk (C.R.); david.collins@mrc-epid.cam.ac.uk (D.C.); polly.page@mrc-epid.cam.ac.uk (P.P.); 2MRC Elsie Widdowson Laboratory, Cambridge CB1 9NL, UK

**Keywords:** sugars, added sugars, free sugars, nutrition survey, UK National Diet and Nutrition Survey, dietary recommendations, public health, children, teenagers, adults

## Abstract

Monitoring dietary intake of sugars in the population’s diet has great importance in evaluating the efficiency of national sugar reduction programmes. The study objective was to provide a comprehensive assessment of dietary sources of added and free sugars to assess adherence to public health recommendations in the UK population and to consider the impact of different sugar definitions on monitoring. The terms “added sugar” and “free sugar” are different sugar definitions which include different sugar components and may result in different sugar intakes depending on the definition. Dietary intake of added sugars, free sugars and seven individual sugar components (sugar from table sugar; other sugars; honey; fruit juice; fruit puree; dried fruit; and stewed fruit) of 2138 males and females (1.5–64 years) from the National Diet and Nutrition Survey (NDNS) 2014–2016, collected using a 4 day estimated food diary, were studied. Added and free sugar intake accounted for 7% to 13% of total energy intake respectively. Major sources of free sugar intake were “cereals and cereal products”, “non-alcoholic beverages”, and “sugars, preserves, confectionery”. Differences between added and free sugar intake were significantly large, and thus use of free sugar versus added sugar definitions need careful consideration for standardised monitoring of sugar intake in relation to public health.

## 1. Introduction

Dietary risk factors are one of the most common causes of non-communicable diseases. Poor diet is also a risk factor for obesity, which is a rapidly increasing independent risk factor for many non-communicable diseases worldwide [1]. High intake of free sugars is a public health concern as it is associated with poor diet, obesity and risk of non-communicable diseases [1]. Dietary recommendations are presently among the most common measures that governments and health organisations use to monitor sugar intake [2]. Increasingly, countries around the world are introducing measures like recommendations [2], reformulation programmes (e.g., Norwegian action plan for a healthier diet [3]) and tax on sugar-sweetened beverages [4] (e.g., tax to sweetened sugar beverages in Mexico [5] and Berkeley, California [6]) to monitor and reduce sugar intake. In 2015, the UK moved from monitoring sugars in terms of non-milk extrinsic sugars (NMES) to monitoring free sugar intakes [7]. Since 2016, the UK Government has been leading a plan for action to reduce childhood obesity [8]. Part of this includes a sugar reduction and wider reformulation programme introduced by Public Health England [9] and a “sugar tax” on sweetened drinks, officially called the Soft Drinks Industry Levy which took effect in April 2018 [10]. Efficient quantification of the dietary sources of added and free sugar in the population’s diet has prominent importance in evaluating sugar reduction programmes. The World Health Organization (WHO) issued dietary guidelines which recommend limiting free sugar intake to less than 10% of daily energy intake [1]. In the UK, the recommendation by the Scientific Advisory Committee on Nutrition (SACN) is for no more than 5% of total energy intake to come from free sugars [7]. However, there are different and inconsistent definitions of what constitutes added and free sugars that are used by international and national organisations [11] which present important considerations for public health monitoring and surveillance programmes and for comparison across countries and over time. For example, the free sugar definition used by the SACN in the UK [7] includes sugars within pureed fruits, whereas this distinction is not made by the WHO [1]. 

The term “sugars,” as applied to human diets, is a collective term for several different chemical species. Thus, “table sugar” is essentially pure sucrose, whereas fruit juice, honey and syrups contain mixtures of sucrose, glucose and fructose, and often oligosaccharides of different size. These compounds are invariably combined as “sugars”. In this paper, added sugars are defined according to the European Food Safety Authority (EFSA) definition [12] which includes sucrose, fructose, glucose, starch hydrolysates (glucose syrup, high-fructose syrup) and other isolated sugar preparations which are added during food preparation and manufacturing [12]. The added sugar definition does not include sugars present in unsweetened fruit juice or honey [7]. The WHO definition [1] for free sugar includes all sugars that are added during food manufacturing and preparation as well as sugars that are naturally present in honey, syrups, fruit juices and fruit concentrates, whereas, in addition, the SACN free sugar definition [7] includes the sugar derived from pureed fruits. For clarity, although not estimated here, the definition of NMES includes sugars not contained within the cellular walls of plants, all sugars added to foods and 50% of the sugars in canned, stewed, dried or preserved fruits [13].

Added sugars and free sugars encompass various components such as table sugar, honey and syrups which are usually added to foods as sweeteners [7]. The measurement of added and free sugar content of foods and drinks is a challenge as there is no laboratory based analytical method or biomarker [14] that can objectively measure these sugars due to the difficulty in distinguishing intrinsic sugars from extrinsic sugars [15,16]. There are some methods [15,17] which estimate the added and free sugars content of foods in a systematic way from intake data. However, these are mostly specific to a single definition such as added sugar or free sugar only and are difficult to use more flexibly. We therefore previously developed a methodology [18] to estimate sugar intake in accordance with the range of different definitions for sugars in order to provide a mechanism to systematically estimate population intakes with the ability to specify and differentiate between the different sugar definitions, for example for total, free, added sugars and NMES, depending on what is desirable to measure. This estimation method [18] can also provide detailed data for seven components of sugars such as sugar from honey and sugar from fruit juice. This method [18] was used to generate data on dietary added and free sugar intake in this study.

Our study aims to present the latest estimation of sugar intakes for the UK population using the most recently available representative population data (pre-introduction of sugar tax) from the National Diet and Nutrition Survey Rolling Programme (NDNS RP). Through this illustration, we also demonstrate the utility of different definitions of sugars, e.g., added sugars and free sugars and consider their impact and relevance to population estimates.

## 2. Materials and Methods

### 2.1. Sugar Definitions Used in This Study

This study used added sugar (defined according to EFSA [12]) and free sugar definitions (defined according to both the WHO [1] and the SACN [7]) to assess the impact of different sugar definitions on the estimation of dietary intake at the population level. The details about these sugar definitions were defined in the introduction section. Data were not assessed according to the NMES definition, which has been used to set and monitor sugar intake guidance for the last 25 years in the UK [13], as this definition was replaced in 2018 with free sugars following the recommendation of the UK SACN [19]. 

Individual sugar components were estimated where relevant to added and free sugar definitions. These were: (1) sugar in table sugar including all sucrose sugars such as granulated sugar used in baking/confectionary and cakes, etc. (Sugar—table); (2) sugar in other sugar-based sweeteners such as fructose, glucose syrup, golden syrup, maple syrup and malt extract (Sugar—other); (3) sugar in honey (Sugar—honey); (4) sugar in fruit and vegetable juice (Sugar—fruit juice); (5) sugar in fruit puree (Sugar—fruit puree). In addition, this study also estimated sugar intakes derived from dried fruit (Sugar—dried fruit) and stewed fruit (Sugar—stewed fruit) as they can be the focus of other public health interests such as risk of dental carries [20,21] or chronic diseases [22]. 

In this study, the composition of sugar components in foods was estimated using our previously published method and the Free Sugar Database [18] to assign sugar sources and values for specific components (e.g., sugar from honey, sugar from fruit juice) to the constituent ingredients of recipes for those foods that included multiple ingredients (e.g., chocolate bar). The intakes of sugar components were summed in combinations according to the three definitions as below: Added sugars (EFSA): Sugar—table + Sugar—other; Free sugars (WHO): Sugar—table + Sugar—other + Sugar—honey + Sugar—fruit juice; Free sugars (SACN): Sugar—table + Sugar—other + Sugar—honey + Sugar—fruit juice + Sugar—pureed fruit.

### 2.2. Estimating Population Intakes of Sugars

The NDNS RP [23] is a continuous running cross-sectional survey assessing the food consumption and nutrient intakes in a nationally representative sample of more than 1000 respondents aged 1.5 years and over, living in private households in the UK each year. Dietary data were collected using a 4 day estimated food diary. Participants were asked to report the amount of food consumed using a combination of portion estimates, including household measures, food photographs and weights from packaging.

This study used data from NDNS RP years 2014–2016 [23] because it provided the most current data available at the time before the introduction of the UK sugar tax (*n* = 2138). In addition, statistically, it was more appropriate to use the combination years 2014–2016, as analysis weights were the most currently available weights to ensure results are generalizable to the UK population at that time. 

Intakes were based on individual average consumption data for the reported four days focused on children (4–10 years) (*n* = 514), teenagers (11–18 years) (*n* = 542) and adults (19–64 years) (*n* = 1082) as these population groups have the largest sample size in the NDNS RP, and their sugar intake coming from discretionary foods was the highest and therefore they were the highest concern for public health. Intakes of pre-school children (1.5–4 years) and older adults (65 years and over) are provided in a supplementary document (Appendix A). 

### 2.3. Statistic Tests 

The statistical programme R version 3.3.2 was used to perform paired t-tests. For the estimation of the added sugars (EFSA) and free sugars (WHO and SACN) consumption from NDNS 2014–2016, the “survey” package within R was used which takes into account the complex survey design (i.e., sample stratification, clustering and weighting) to yield valid estimates of the population parameters [23]. Intake levels are presented as median and 25th, and 75th percentile values (due to the skewed nature of the data). The significance of difference (*p* < 0.05) in intakes estimated using the different definitions free sugars (WHO), free sugars (SACN) and added sugars (EFSA) were tested with Wilcoxon signed-rank test. The significance of difference of sugar intake as a percentage of Total Energy (% TE) between children, teenagers and adults were tested with Mann–Whitney U test. 

## 3. Results

### 3.1. Estimates of Added Sugar and Free Sugar Consumption

Median daily added sugars (EFSA) intakes were 38.5 g (10% of total energy (TE)) for 4–10-year-olds, 49.9 g (11% TE) for 11–18-year-olds and 34.8 g (7% TE) for 19–64-year-olds (Table 1). The median daily intakes of free sugars as defined by the WHO and the SACN respectively were 46.6g (13% TE) and 47.8 g (13% TE) for 4–10-year-olds, 58.7 g (13% TE) and 60.1 g (13% TE) for 11–18-year-olds and 44.8 g (9% TE) and 45.6 g (10% TE) for adults.

In all cases, intakes of sugars were ranked in the following order; added sugars (EFSA) < free sugars (WHO) < free sugars (SACN). These differences are due to the differences in definition and therefore the difference in the sub-categorisation of sugars included in the calculation. The difference between free sugar intake defined by the WHO and defined SACN (as a percent of energy intake) was not statistically significant (*p* < 0.05)**,** ranging from 0.8 g in adults (19 to 64 years) to 1.4 g in teenagers (11 to 18 years), corresponding to a maximum of 1% of total energy only. The small differences observed were attributable to the differences in free sugar definitions: the SACN definition includes sugar derived from fruit puree whereas the WHO definition does not and sugar consumption from fruit puree was very low (median intake ranging from 0.1 g in 19–64-year-olds to 0.3 g in 4–10-year-olds) (Table 2).

The difference between added sugars (EFSA) and free sugars (both the WHO and the SACN definitions) intake as a percent of energy intake was statistically significant (*p* < 0.05), ranging from a difference of 8.1 g in children (4 to 10 years) to 10.8g in adults (19 to 64 years), corresponding to a maximum of 3% of total energy (Table 1). This difference was mainly due to the contribution of sugar from fruit juice, as both the WHO and the SACN free sugar definitions include sugar derived from fruit juice, whereas added sugars (EFSA) does not (Table 2).

Across all age groups, teenagers (11 to 18 years) had the highest added sugars (EFSA) intake (median 49.9 g per day) mainly due to a higher intake of table sugar (all sucrose sugars such as granulated sugar used in baking/confectionary and cakes, etc.) compared to the other age groups (Table 1). The daily intakes of added sugars (EFSA) and free sugars (the WHO and the SACN definitions) as a percentage of energy intake were significantly higher in children (4 to 10 years) and teenagers (11 to 18 years) compared with adults (19 to 64 years), whereas there was no statistical difference between children (4 to 10 years) and teenagers (11 to 18 years). Between male and females, there was no statistical difference in added sugars (EFSA) and free sugars (the WHO and the SACN definitions) as a percentage of energy intakes.

### 3.2. Estimates of Individual Sugar Components

When looking at the sugar components, table sugar was the main contributor to free sugars (WHO/SACN) intake across all population groups and it was higher in population groups with higher free sugar intake (Table 1 and Table 2). For example, the median table sugar intake of teenagers (11 to 18 years) was 43.6 g, whereas the sum of sugars from other sources (e.g., sugar from fruit juice) was 9.6 g. The next main contributors to free sugar intake were other sugars and sugar from fruit juice. The sugar source contributing the least to overall sugar intake was Sugar—honey and Sugar—stewed fruit across all population groups. 

Between 10% and 20% of added sugars (EFSA, which consist of Sugar—table + Sugar—other only) was derived from the Sugar—other component. For example, in adults, the ratio of Sugar—table to Sugar—other intake was 10:1.

When looking at consumers only, for most individuals in the NDNS population sample, sugar consumption was derived from table sugar (100% consumers), other sugars (96% consumers), fruit juice (92% consumers) and pureed fruit (69% consumers) (Table 3). In contrast, fewer individuals had consumed sugar derived from dried fruit (43% consumers), honey (26% consumers) and stewed fruit (27% consumers) (Table 3).

### 3.3. Adherence to Public Health Recommendations for Free Sugar intakes and A Comparison between the WHO and SACN Free Sugar Definitions

This study looked at the proportion who met the free sugar dietary recommendations by the WHO (no more than 10% of total energy should be derived from free sugars) [1] and the SACN [7] (no more than 5% of total energy should be derived from free sugars) (Table 4). A greater proportion of the population met the WHO recommendation than the more stringent SACN recommendation which is to be expected. The proportion of people meeting the WHO recommendation ranged between 25% and 54%, whereas the proportion meeting the SACN recommendation ranged between 3% and 15%. The percentage meeting recommendations were higher in adults (19 to 64 years) compared with teenagers (11 to 18 years) and children (4 to 10 years).

This was the case whether free sugar intakes were calculated in accordance with the WHO or the SACN definitions for what constitutes “free sugars”. Differences in proportions meeting the recommendations were very minimal or non-existent between the WHO and the SACN definitions in all age groups (see Table 4).

### 3.4. Contribution of Food Sources to Sugar Intakes

In Table 5, Table 6 and Table 7, foods are organized into food groups and subgroups as reported in the NDNS RP and presented in relation to their contribution to intake of sugars for the different definitions (i.e., added sugars, EFSA; free sugars, the WHO and the SACN) and to specific sugar components (i.e., Sugar—honey, Sugar—fruit juice, Sugar—pureed fruit, Sugar—stewed fruit and Sugar—dried fruit)); only NDNS RP food groups which contributed to sugar intake are presented.

In all age groups, “cereal and cereal products”, “non-alcoholic beverages”, “sugars, preserves, confectionery” and “milk and milk products” contributed most to the intake of both added sugars (EFSA) and free sugars (the WHO and the SACN definitions), for example for teenagers the total contribution from these food groups ranged from 91% to 92%. In teenagers, the largest contribution to free sugar intake (33%), was from “non-alcoholic beverages”, mostly derived from sugary soft drinks (23%) including 10% contribution of “fruit and vegetable juices”. The highest contributor of free sugar intake was “cereals and cereal products” for children (4 to 10 years) (33%–34%) and “sugar, preserves and confectionary” for adults (19 to 64 years) (26%).

In all age groups, the main contribution of “cereal and cereal products”, “non-alcoholic beverages”, “sugars, preserves, confectionery” and “milk and milk products” was derived from their added sugar (EFSA) (Sugar—table + Sugar—other) content. However, these food groups also contribute to the intake of specific sugar components such as sugars-fruit juice. For example, in all age groups, “sugars, preserves, sweet spreads” contributed to 8%–12% of Sugar—pureed fruit intake, and, in children and teenagers, “sugar confectionary” contributed to 6%–10% of Sugar—fruit juice intake. Specific sugar components such as Sugar—honey can also add into free sugar intake through foods that can be seen as healthy or sugar free [24,25]. For example, across all age groups, “breakfast cereals” contributed to Sugar—honey intake (7%–14%), “yogurt, fromage frais, dairy desserts” (12%–25%) and “diet soft drinks” (8%–21%) contributed to the Sugar—pureed fruit intake, and “savoury sauces and condiments” contributed to 3%–6% of Sugar—fruit juice intake which indicates that honey is widely used in breakfast cereals, pureed fruit is used in “yogurt, fromage frais, dairy desserts” and “diet soft drinks”, and fruit juice is used in “savoury sauces and condiments”.

## 4. Discussion

In the UK NDNS population (4–64 years old) (2014–2016) the estimated daily intake of added sugars (EFSA) ranged from 7% TE to 11% TE. These results are similar to the figures presented in a review [26] of added sugar consumption reported in nine nutrition surveys across the world (6% TE to 19% TE in people over 4 years old). In the present study, the intake of free sugars as defined by the WHO and the SACN ranged from 9% TE to 13% TE, similar to the free sugar intake reported in the Dutch National Food Consumption Survey 2007–2010 [16], in the New Zealand Adult Nutrition Survey 2008/09 [17] and in Swiss National Nutrition Survey (2014–2015) [27] which were, on average, 14% TE in 7–69 years old, 11% TE in 15–71 years old and 11% TE in 18–75 years old, retrospectively.

Intakes of added and free sugar were higher in children and teenagers compared with adults. This was also found in the review of added sugar consumption reported in nine nutrition surveys across the world [26] (up to 19% TE in adolescents and 16% TE in adults) and a review [28] of surveys in European countries (11 to 17% TE in children and adolescents and 7 to 11% TE in adults). In this present study, the proportion of people meeting the WHO and the SACN free sugar recommendations was relatively low in all population groups, and overall adults showed more adherence to recommendations than teenagers and children. Likewise, in the Dutch National Food Consumption Survey (2007–2010) [16], 5% of teenagers (7–18 years) and 29% to 33% of adults (19–69 years) met the WHO free sugar guidelines. In the Switzerland National Nutrition Survey (2014–2015) [27], 36%, 45%, and 53% of people aged 18–29, 30–64, and 65–75 years, respectively, met the WHO free sugar guidelines. The results of a meta-analysis [29] also indicated an overall decrease in added sugar intake from adolescence to early adulthood. This present study hasn’t looked at the trends in free sugar intake in the UK since this was not the particular focus of this paper and a trend analysis was published in an NDNS report on Years 1–9 results [30]. In summary, NDNS Years 1-9 results [31] has shown a reduction in free sugar intake over a 9 year period (2008–2017) in children, teenagers and adults, for example there was an average yearly reduction of 0.3% and 0.4% TE percentage points in 1.5 to 10 years and 11 to 18 years. The results [30] also showed a downward trend in the proportion of children and teenagers consuming sugar-sweetened soft drinks and, for consumers, a reduction the amount drank per day (e.g., a drop from approx. 285 g/day to 185 g/day among teenage consumers over the 9 years). Although these findings are promising, the dietary sugar intake in children and teenagers still remains particular public health concern. The UK Government’s long-term initiatives to reduce sugar intake includes the Change4Life public health campaign [32] and the sugar reduction and wider reformulation programme [9]. The ongoing survey will be instrumental in identifying sugar rich foods and enable continuous monitoring of the impact of these initiatives.

The difference between intake of added sugars (EFSA) and free sugars (both the WHO and the SACN definitions) was statistically significant (e.g., for teenagers, median intake of added sugars and free sugars was 11% TE and 13% TE (*p* < 0.05), respectively), which was mainly due to the sugar contribution from fruit juice. This study also showed that, for children and teenagers, discretionary food groups like “sugars, preserves, sweet spreads” and “sugar confectionary” contributed to Sugar—honey, Sugar—fruit juice and Sugar—pureed fruit intake (2%–15.6%). This indicates that discretionary sugar-rich foods are not only a concern for their added sugar content but also a concern for their concentration of sugars coming from natural sources (sugars contained within the cellular walls of plants such as sugars in fruit juice and honey). This shows the efficiency of using a free sugar definition as opposed to added sugar, since free sugar definition covers sugar derived from a wider range of food sources like fruit juices. Due the differences between added sugar and free sugar definitions, future studies may need careful consideration on using free sugar or added sugar definitions for the assessment of sugar intake; this may have implications for public health policy and initiatives.

The WHO definition [1] for free sugars includes all sugars that are added during food manufacturing and preparation and are naturally present in honey, syrups, fruit juices and fruit concentrates, whereas, in addition, the SACN free sugar definition [7] includes the sugar derived from pureed fruits. In this study the median Sugar—fruit puree intake was not very different when looked into for consumers of Sugar—fruit puree only (0.3–0.4 g/day) compared with for consumers and non-consumers of Sugar—fruit puree combined (0.1–0.3 g/day). The difference between free sugar intake as defined by the WHO and the SACN was not statistically different. This suggests that the inclusion of sugar contribution from fruit puree into a free sugar definition has minimal impact on population level intake estimates. However, the individual level intake estimates could vary according to how much fruit puree consumers eat.

A review [32] published in 2014 showed a variability across national nutrition surveys in terms of the sugar definitions use which included “total sugars”, “non-milk extrinsic sugars”, “added sugars”, “sucrose” and “mono- and disaccharides”. A later review [26] published in 2016 collated data from nationally representative nutrition surveys and showed that only a few surveys reported intakes of added sugars whereas no country reported intakes of free sugars. The variation in the use of sugar definitions was shown as a limiting factor for comparisons to be made across countries; therefore, a consistent and uniform approach to the estimation of dietary sugar intake national nutrition surveys was recommended [26,32].

The highest source of sugar intake was derived from Sugar—table ingredients (6% TE –10% TE) such as granulated table sugar consumed alone or as a sweetener in processed foods and drinks. Some added sugar (EFSA) (1% TE) was also derived from Sugar—other ingredients which are likely to be derived from hidden sugar ingredients such as isoglucose. These hidden ingredients can be challenging to identify as added sugar on food labels [24,33]. This reinforces the need for clear labelling of ingredients in foods alongside product reformulation programmes.

There are two limitations in this study. Firstly, added and free sugar values of foods are derived from an estimated method [18] as there is no analytical method to determine actual added sugars and free sugars content of foods. However, it is also strength of this study that the free sugar estimation method [18] has been validated and followed a very comprehensive approach which included the disaggregation of complex foods. Like in all other surveys, the misreporting of foods high in sugar could affect the estimates of intake in this study. As a generally accepted limitation of subjective dietary assessment methods [34], misreporting can differentiate the results as individuals may alter their dietary pattern or misreport their food and drink intake, especially through underreporting foods perceived as unhealthy [35]. Therefore, it is possible that the dietary intake of added and free sugar intake may be higher than reported here. The NDNS RP assesses misreporting through the Doubly-Labelled Water method the results of which are available elsewhere [36].

This study provides the most recent assessment of added sugar and free sugar intake (pre-introduction of sugar tax) across all age and sex groups in the UK from age 1.5 years and above, using nationally representative data. Utilising a detailed systematic calculation method, for which added sugars and free sugars are split into their constituent parts (such as sugar derived from honey or fruit juice) enabled a comprehensive quantification of sugar intake. Through the illustration of intakes in relation to recommended intakes in accordance to different definitions of sugars, this study compares the utility and benefit of using dietary intake values derived for different definitions of sugar intake, namely added sugar (EFSA), free sugar (WHO) and free sugar (SACN, UK). This study and analysis can help future researchers and public health programmes to assess and understand the implications of applying different definitions, the importance of being systematic in the assessment of intake and provides detail to enable replication and thus comparison between other studies or for use in country specific nutrition surveys. In the next few years following the sugar tax, the results of this study can also be useful to evaluate the impact of UK sugar tax.

## 5. Conclusions

In the UK, the proportion of people (4 to 64 years) meeting the WHO free sugar recommendation ranged between 25% and 54%, whereas the proportion meeting the SACN free sugar recommendation ranged between 3% and 15%. Across all age groups, added and free sugar intake accounted for 7% to 13% of total energy intake. The highest source of sugar intake derived from sugar rich ingredients such as granulated sugar and glucose syrup which are used in baking/confectionary and cakes, etc. The major sources of free sugars were “cereals and cereal products”, “non-alcoholic beverages”, “sugars, preserves, confectionery” and “milk and milk products”. Sugar sweetened soft drinks were the highest contributor of free sugar intake among teenagers. The difference between added sugars (EFSA) and free sugars (both the WHO and the SACN definitions) intake was significantly large and estimates of free sugars and added sugars cannot be considered directly comparable. Free sugars provides for a more comprehensive assessment of sugar intake for population nutritional surveillance. There was little difference between the detail of definitions of free sugars by the WHO and the SACN suggesting data compiled according to either or similar definitions are comparable. Studies need careful consideration as decision on which definition to use, in relation to whether to estimate intakes of free sugars or added sugars for the assessment of sugar intake in study populations.

## Figures and Tables

**Table 1 nutrients-12-00393-t001:** Daily intakes of added and free sugars by age and sex (NDNS 2014–2016) (including consumers and non-consumers).

Added and Free Sugars Combinations		Children, 4–10 Years			Teenagers, 11–18 Years			Adults, 19–64 Years		
		Male (*n* = 276)	Female (*n* = 238)	Combined (*n* = 514)	Male (*n* = 270)	Female (*n* = 272)	Combined (*n* = 542)	Male (*n* = 450)	Female (*n* = 632)	Combined (*n* = 1082)
Added sugars (EFSA) ^1^	g/day, Median	40.2	36.9	38.5	50.5	49.0	49.9	38.2	32.3	34.8
	g/day, P25–P75	25.9–55.4	27.0–50.9	26.9–53.1	32.5–72.3	27.3–68.4	30.1–72.4	21.3–61.7	19.0–51.5	19.3–56.6
	% TE, Median	10 ^a,b^	11 ^a,b^	10 ^a,b^	11 ^a,b^	12 ^a,b^	11 ^a,b^	7 ^a,b,c,d^	8 ^a,b,c,d^	7 ^a,b,c,d^
	% TE, P25–P75	7–13	8–13	7–13	7–15	8–15	8–15	4–11	5–12	4–11
Free sugars (WHO) ^2^	g/day, Median	49.0	43.6	46.6	62.9	55.8	58.7	52.0	40.9	44.8
	g/day, P25–P75	33.7–65.4	30.7–58.7	32.9–63.1	41.4–83.8	31.2–78.3	37.5–79.2	29.6–78.5	24.7–64.1	27.0–72.4
	% TE, Median	13	13	13	13	13	13	9 ^c,d^	10 ^c,d^	9 ^c,d^
	% TE, P25–P75	9–16	10–16	9–16	9–17	9–18	9–17	6–14	6–14	6–14
Free sugars (SACN) ^3^	g/day, Median	50.3	44.5	47.8	63.6	57.1	60.1	52.5	41.6	45.6
	g/day, P25–P75	34.3–66.2	31.7–60.4	33.9–64.1	41.6–85.4	31.4–78.7	37.5–80.3	30.1–79.9	25.5–64.6	27.1–73.0
	% TE, Median	13	13	13	13	13	13	9 ^c,d^	10 ^c,d^	10 ^c,d^
	% TE, P25–P75	9–16	10–16	10–16	10–17	10–18	10–18	6–14	6–14	6–14

NDNS: UK National Diet and Nutrition Survey. *n*= number of participants. ^1^Added sugars (European Food Safety Authority, EFSA): Sugar—table + Sugar—other; ^2^Free sugars (World Health Organisation, WHO): Added sugars (EFSA) + Sugar—honey + Sugar—fruit juice; ^3^Free sugars (UK Scientific Advisory Committee on Nutrition, SACN): Free sugars (WHO) + Sugar—pureed fruit; P25-P75: Inter-Quartile Range (percentile 25^th^–75^th^). ^a^ Significant difference between free sugars (WHO) and added sugars (EFSA) (*p* < 0.05). ^b^ Significant difference between free sugars (SACN) and added sugars (EFSA) (*p* < 0.05). ^c^ Significant difference between % Total Energy (TE) of adults and children (*p* < 0.05). ^d^ Significant difference between% TE of adults and teenagers (*p* < 0.05).

**Table 2 nutrients-12-00393-t002:** Daily intakes of individual sugar components by age and sex (NDNS 2014–2016) (including consumers and non-consumers).

Individual Sugar Components		Children, 4–10 Years			Teenagers, 11–18 Years			Adults, 19–64 Years		
		Male (*n* = 276)	Female (*n* = 238)	Combined (*n* = 514)	Male (*n* = 270)	Female (*n* = 272)	Combined (*n* = 542)	Male (*n* = 450)	Female (*n* = 632)	Combined (*n* = 1082)
Sugar—table ^1^	g/day, Median	33.4	30.8	32.5	45.2	41.3	43.6	32.9	28.0	30.4
	g/day, P25–P75	21.9–46.7	23.4–42.9	22.8–44.9	28.1–62.8	25.0–59.4	26.6–61.1	17.3–56.0	15.7–44.1	16.6–50.5
	% TE, Median	9	9	9	10	10	10	6	7	6
	% TE, P25–P75	6–11	7–11	6–11	6–13	7–13	7–13	3–10	4–10	4–10
Sugar—other ^2^	g/day, Median	5.2	4.8	5.1	4.7	4.6	4.6	3.1	2.9	3.0
	g/day, P25–P75	2.7–9.5	2.6–8.4	2.7–9.2	1.4–10.9	1.4–11.8	1.4–11.3	0.7–6.8	0.9–6.3	0.8–6.5
	% TE, Median	1	1	1	1	1	1	1	1	1
	% TE, P25–P75	1–2	1–2	1–2	0–2	0–2	0–2	0–1	0–1	0–1
Sugar—honey ^3^	g/day, Median	0.0	0.0	0.0	0.0	0.0	0.0	0.0	0.0	0.0
	g/day, P25–P75	0.0–0.1	0.0–0.0	0.0–0.1	0.0–0.0	0.0–0.0	0.0–0.0	0.0–0.0	0.0–0.1	0.0–0.1
	% TE, Median	0	0	0	0	0	0	0	0	0
	% TE, P25–P75	0–0	0–0	0–0	0–0	0–0	0–0	0–0	0–0	0–0
Sugar—fruit juice ^4^	g/day, Median	4.8	4.4	4.6	6.1	3.6	4.9	4.8	3.3	3.9
	g/day, P25–P75	1.1–13.3	0.7–10.4	0.9–11.4	0.5–13.8	0.6–11.1	0.6–12.5	0.2–18.1	0.3–9.8	0.3–12.9
	% TE, Median	1	1	1	1	1	1	1	1	1
	% TE, P25–P75	0–3	0–3	0–3	0–3	0–2	0–3	0–3	0–2	0–3
Sugar—pureed fruit ^5^	g/day, Median	0.3	0.3	0.3	0.1	0.1	0.1	0.1	0.1	0.1
	g/day, P25–P75	0.1–0.8	0.1–0.7	0.1–0.8	0.0–0.4	0.0–0.4	0.0–0.4	0.0–0.4	0.0–0.5	0.0–0.5
	% TE, Median	0	0	0	0	0	0	0	0	0
	% TE, P25–P75	0–0	0–0	0–0	0–0	0–0	0–0	0–0	0–0	0–0
Sugar—stewed fruit ^6^	g/day, Median	0.0	0.0	0.0	0.0	0.0	0.0	0.0	0.0	0.0
	g/day, P25–P75	0.0–0.0	0.0–0.0	0.0–0.0	0.0–0.0	0.0–0.0	0.0–0.0	0.0–0.0	0.0–0.1	0.0–0.1
	% TE, Median	0	0	0	0	0	0	0	0	0
	% TE, P25–P75	0–0	0–0	0–0	0–0	0–0	0–0	0–0	0–0	0–0
Sugar—dried fruit ^7^	g/day, Median	0.0	0.0	0.0	0.0	0.0	0.0	0.0	0.1	0.0
	g/day, P25–P75	0.0–2.2	0.0–0.8	0.0–1.7	0.0–0.2	0.0–0.7	0.0–0.5	0.0–2.4	0.0–2.9	0.0–2.7
	% TE, Median	0	0	0	0	0	0	0	0	0
	% TE, P25–P75	0–0	0–0	0–0	0–0	0–0	0–0	0–0	0–1	0–1

NDNS: UK National Diet and Nutrition Survey. *n*= number of participants. ^1^ Sugar—table: Including all sucrose sugars such as granulated sugar used in baking/confectionary and cakes, etc.; ^2^ Sugar—other: Sugar in other sugar-based sweeteners such as glucose syrup; ^3^ Sugar—honey: Sugar in honey; ^4^ Sugar—fruit juice: Sugar in fruit and vegetable juice; ^5^ Sugar—pureed fruit: Sugar in fruit puree; ^6^ Sugar—stewed fruit: Sugar in stewed fruit; ^7^ Sugar—dried fruit; Sugar in dried fruit; % TE: total energy. P25–P75: Inter-Quartile Range (percentile 25^th^–75^th^).

**Table 3 nutrients-12-00393-t003:** Daily intakes of individual sugar components by age and sex (NDNS 2014–2016) (consumers only).

Individual Sugar Components		Children, 4–10 Years			Teenagers, 11–18 Years			Adults, 19–64 Years		
		Male	Female	Combined	Male	Female	Combined	Male	Female	Combined
Sugar—table ^1^	% Consumers	100	100	100	100	100	100	100	100	100
	g/day, Median	33.4	30.8	32.5	45.2	41.3	43.6	33.0	28.0	30.5
	g/day, P25–P75	21.9–46.7	23.4–42.9	22.8–44.9	28.1–62.8	25.0–59.4	26.6–61.1	17.6–56.1	15.8–44.2	16.8–51.3
	% TE, Median	9	9	9	10	10	10	6	7	6
	% TE, P25–P75	6–11	7–11	6–11	6–13	7–13	7–13	4–10	4–10	4–10
Sugar—other ^2^	% Consumers	100	98	99	97	97	97	94	95	95
	g/day, Median	5.2	4.9	5.2	5.2	4.8	4.9	3.3	3.2	3.3
	g/day, P25–P75	2.7–9.7	2.7–8.4	2.7–9.2	1.7–11.2	1.5–11.9	1.5–11.7	1.0–7.2	1.2–6.7	1.1–7.1
	% TE, Median	1	1	1	1	1	1	1	1	1
	% TE, P25–P75	1–2	1–2	1–2	0–2	0–3	0–2	0–1	0–1	0–1
Sugar—honey ^3^	% Consumers	30	27	28	23	22	23	24	28	26
	g/day, Median	0.3	0.4	0.4	0.3	0.3	0.3	0.9	1.3	1.1
	g/day, P25–P75	0.1–1.6	0.1–1.6	0.1–1.6	0.1–2.0	0.1–0.5	0.1–1.4	0.3–4.8	0.2–4.0	0.3–4.5
	% TE, Median	0	0	0	0	0	0	0	0	0
	% TE, P25–P75	0–0	0–0	0–0	0–0	0–0	0–0	0–1	0–1	0–1
Sugar—fruit juice ^4^	% Consumers	98	96	97	90	93	92	91	90	90
	g/day, Median	5.2	4.7	4.8	7.1	4.3	5.5	6.8	4.1	4.8
	g/day, P25–P75	1.3–13.8	0.9–10.5	1.2–11.7	1.7–16.5	0.8–12.2	1.0–12.9	1.2–19.8	0.8–10.8	0.8–14.1
	% TE, Median	1	1	1	1	1	1	1	1	1
	% TE, P25–P75	0–3	0–3	0–3	0–3	0–3	0–3	0–3	0–2	0–3
Sugar—pureed fruit ^5^	% Consumers	87	89	88	61	65	63	60	67	64
	g/day, Median	0.4	0.3	0.4	0.3	0.3	0.3	0.3	0.3	0.3
	g/day, P25–P75	0.2–1.1	0.2–0.8	0.2–1.0	0.1–0.8	0.1–0.7	0.1–0.8	0.1–0.9	0.1–0.8	0.1–0.8
	% TE, Median	0	0	0	0	0	0	0	0	0
	% TE, P25–P75	0–0	0–0	0–0	0–0	0–0	0–0	0–0	0–0	0–0
Sugar—stewed fruit ^6^	% Consumers	23	27	25	22	25	23	26	32	30
	g/day, Median	0.2	0.3	0.3	0.2	0.2	0.2	0.4	0.2	0.3
	g/day, P25–P75	0.1–0.7	0.1–1.1	0.1–0.9	0.1–0.7	0.0–0.4	0.1–0.5	0.1–1.4	0.1–0.8	0.1–1.2
	% TE, Median	0	0	0	0	0	0	0	0	0
	% TE, P25–P75	0–0	0–0	0–0	0–0	0–0	0–0	0–0	0–0	0–0
Sugar—dried fruit ^7^	% Consumers	42	40	41	28	37	32	45	51	49
	g/day, Median	2.4	1.3	2.0	1.4	1.3	1.3	3.0	2.6	2.8
	g/day, P25–P75	0.12-0.4	0.0–17.8	0.1–18.6	0.0–17.8	0.0–19.4	0.0–19.4	0.0–34.1	0.0–20.3	0.0–25.1
	% TE, Median	1	0	1	0	0	0	1	1	1
	% TE, P25–P75	0–5	0–4	0–5	0–2	0–5	0–5	0–6	0–5	0–5
Bases (unweighted)	n, consumers and non–consumers	276	238	514	270	272	542	450	632	1082

NDNS: UK National Diet and Nutrition Survey. ^1^ Sugar—table: Including all sucrose sugars such as granulated sugar used in baking/ confectionary and cakes, etc.; ^2^ Sugar—other: Sugar in other sugar-based sweeteners such as glucose syrup; ^3^ Sugar—honey: Sugar in honey; ^4^ Sugar—fruit juice: Sugar in fruit and vegetable juice; ^5^ Sugar—pureed fruit: Sugar in fruit puree; ^6^ Sugar—stewed fruit: Sugar in stewed fruit; ^7^ Sugar—dried fruit; Sugar in dried fruit; % TE: total energy.% Consumers: Percentage of consumers of this type of sugar; P25-P75: Inter-Quartile Range (percentile 25^th^- 75^th^). n= number of participants.

**Table 4 nutrients-12-00393-t004:** Percentage meeting recommendations for free sugar intake by sex and age groups (NDNS 2014–2016) (including consumers and non-consumers).

Added and Free Sugars Combinations		%								
	Recommendation Threshold	Children, 4–10 Years			Teenagers, 11–18 Years			Adults, 19–64 Years		
		Male (*n* = 276)	Female (*n* = 238)	Combined (*n* = 514)	Male (*n* = 270)	Female (*n* = 272)	Combined (*n* = 542)	Male (*n* = 450)	Female (*n* = 632)	Combined (*n* = 1082)
Added sugars (EFSA) ^1^	<5% TE ^4^	8	5	7	10	9	10	31	28	29
	<10% TE ^5^	49	42	46	41	38	39	69	68	69
Free sugars (WHO) ^2^	<5% TE ^4^	3	3	3	6	6	6	15	15	15
	<10% TE ^5^	32	28	30	28	27	27	54	53	54
Free sugars (SACN) ^3^	<5% TE ^4^	3	3	3	6	6	6	15	15	15
	<10% TE ^5^	29	26	28	27	25	26	54	52	53

NDNS: UK National Diet and Nutrition Survey. n= number of participants. ^1^ Added sugars (European Food Safety Authority, EFSA): Sugar—table + Sugar—other; ^2^ Free sugars (World Health Organisation, WHO): Added sugars (EFSA) + Sugar—honey + Sugar—fruit juice; ^3^ Free sugars (UK Scientific Advisory Committee on Nutrition, SACN): Free sugars (WHO) + Sugar—pureed fruit. ^4^ Recommendation by the SACN (<5% TE: Below 5% of total energy); ^5^ Recommendation by the WHO (<10% TE: Below 10% of total energy).

**Table 5 nutrients-12-00393-t005:** Contribution (%) of NDNS RP food group sources to intake of sugars by age (4–10 Years) (NDNS 2014–2016)*.

	Children, 4–10 Years							
	Added Sugars (EFSA) ^1^	Free Sugars (WHO) ^2^	Free Sugars (SACN) ^3^	Sugar—Honey	Sugar—Fruit Juice	Sugar—Pureed Fruit	Sugar—Stewed Fruit	Sugar—Dried Fruit
Cereals and cereal products	40.4	34.0	33.4	19.4	2.3	9.0	14.7	23.0
Breakfast cereals	9.3	7.9	7.7	13.5	0.0	0.2	0.8	4.5
Biscuits, buns, cakes, pastries, fruit pies	27.8	23.5	23.2	4.7	1.6	8.7	13.9	17.1
Pasta, rice, bread and other cereals	3.3	2.6	2.5	1.2	0.7	0.0	0.0	1.4
Sugars, preserves, confectionery	26.3	23.9	23.9	7.0	10.3	15.6	0.0	2.5
Sugars ^4^, preserves, sweet spreads	9.1	8.4	8.3	6.9	0.0	12.1	0.0	0.0
Sugars consumed in tea and coffee	1.1	1.0	1.0	0.0	0.0	0.0	0.0	0.0
Sugar confectionery	7.1	6.9	7.0	0.0	10.3	3.5	0.0	1.7
Chocolate confectionery	9.0	7.7	7.6	0.1	0.0	0.0	0.0	0.8
Milk and milk products	14.7	12.7	12.7	1.2	1.5	27.8	1.1	0.0
Yogurt, fromage frais, dairy desserts	7.0	6.0	6.1	1.2	0.0	24.6	1.1	0.0
Ice cream	6.2	5.3	5.2	0.0	0.1	3.2	0.0	0.0
Cream and other milks ^5^	1.6	1.5	1.4	0.0	1.7	0.0	0.0	0.0
Fruits and fruit products	0.3	0.3	0.4	0.0	1.2	3.4	4.1	17.0
Non-alcoholic beverages	10.3	22.1	22.6	0.0	76.9	28.1	2.4	0.0
Fruit and vegetable juice	0.0	10.6	10.9	0.0	42.1	6.1	0.0	0.0
Soft drinks ^6^ (not diet ^7^)	10.3	10.2	10.1	0.0	16.0	1.2	2.4	0.0
Soft drinks ^6^ (diet ^7^)	0.0	1.3	1.7	0.0	18.7	20.8	0.0	0.0
Alcoholic beverages	0.0	0.0	0.0	0.0	0.0	0.0	0.0	0.0
Miscellaneous	5.1	4.4	4.4	0.0	2.6	2.2	1.2	0.2
Beverages dry weight ^8^	2.4	2.0	1.9	0.0	0.0	0.0	0.0	0.0
Soups	0.6	0.5	0.5	0.0	0.1	0.0	0.0	0.0
Savoury sauces and condiments	2.1	1.8	1.8	0.0	2.5	1.3	1.0	0.2
Bases (unweighted)	514	514	514	514	514	514	514	514
% consumers	100	100	100	30	97	86	25	44

NDNS: UK National Diet and Nutrition Survey. ^1^ Added sugars (European Food Safety Authority, EFSA): Sugar—table + Sugar—other; ^2^ Free sugars (World Health Organisation, WHO): Added sugars (EFSA) + Sugar—honey + Sugar—fruit juice; ^3^ Free sugars (UK Scientific Advisory Committee on Nutrition, SACN): Free sugars (WHO) + Sugar—pureed fruit. ^4^ Discreationary sugars added to foods and drinks, apart from tea and coffee; ^5^ Other milks include milkshake, coffee whitener, milk drinks, hot chocolate, milk alternatives, dried milk and milks other than cows’ milk. ^6^ Soft drinks include squashes, cordials, tonic water, energy drinks, all types of still and carbonated soft drinks and ice tea. ^7^ Diet refers to low-calorie, no-added-sugar (EFSA) and sugar-free varieties. ^8^ Beverages dry weight includes instant and powder forms of drinking chocolate, cocoa, malted drinks, milk shake powder, tea and coffee such as cappuccino, latte and mocha with sugar. *Some food groups such as meats are removed as they don’t have relevance to this table. Therefore, the sum of % contributions may not always add up to % consumers.

**Table 6 nutrients-12-00393-t006:** Contribution (%) of NDNS RP food group sources to intake of sugars by age (11–18 Years) (NDNS 2014–2016)*.

	Teenagers, 11–18 Years (*n* = 542)							
	Added Sugars (EFSA) ^1^	Free Sugars (WHO) ^2^	Free Sugars (SACN) ^3^	Sugar—Honey	Sugar—Fruit Juice	Sugar—Pureed Fruit	Sugar—Stewed Fruit	Sugar—Dried Fruit
Cereals and cereal products	34.2	29.4	29.1	18.4	2.2	10.1	13.0	23.9
Breakfast cereals	9.1	7.7	7.6	10.2	0.0	0.2	0.3	6.3
Biscuits, buns, cakes, pastries, fruit pies	20.9	18.2	18.0	7.3	1.7	9.9	11.9	16.2
Pasta, rice, bread and other cereals	4.2	3.6	3.5	0.8	0.5	0.0	0.8	1.4
Sugars, preserves, confectionery	24.5	22.3	22.1	5.6	6.0	8.0	0.0	1.9
Sugars ^4^, preserves, sweet spreads	6.5	5.8	5.8	5.4	0.0	7.6	0.0	0.0
Sugars consumed in tea and coffee	4.9	4.4	4.4	0.0	0.0	0.0	0.0	0.0
Sugar confectionery	4.5	4.4	4.4	0.0	5.9	0.3	0.0	0.5
Chocolate confectionery	8.6	7.6	7.5	0.2	0.1	0.0	0.0	1.4
Milk and milk products	8.0	7.1	7.1	0.1	2.4	15.2	1.8	0.1
Yogurt, fromage frais, dairy desserts	2.9	2.6	2.6	0.1	0.0	12.3	1.8	0.1
Ice cream	3.0	2.6	2.6	0.0	0.1	2.6	0.0	0.0
Cream and other milks ^5^	2.1	1.9	1.9	0.0	2.5	0.3	0.0	0.0
Fruits and fruit products	0.0	0.1	0.2	0.0	0.4	3.4	1.9	6.0
Non-alcoholic beverages	24.5	33.2	33.6	0.0	67.7	22.6	3.0	0.0
Fruit and vegetable juice	0.1	9.9	10.1	0.0	39.0	4.3	0.0	0.0
Soft drinks ^6^ (not diet ^7^)	24.3	22.9	22.7	0.0	18.9	0.9	3.0	0.0
Soft drinks ^6^ (diet ^7^)	0.0	0.4	0.7	0.0	9.8	17.4	0.0	0.0
Alcoholic beverages	0.3	1.0	1.0	0.0	4.0	0.0	0.0	0.0
Miscellaneous	4.3	3.5	3.5	0.4	5.9	2.0	1.8	1.3
Beverages dry weight ^8^	1.3	1.1	1.0	0.0	0.0	0.0	0.0	0.0
Soups	0.2	0.2	0.1	0.0	0.0	0.0	0.0	0.1
Savoury sauces and condiments	2.7	2.2	2.2	0.4	5.9	2.0	1.8	1.2
Bases (unweighted)	542	542	542	542	542	542	542	542
% consumers	100	100	100	26	92	62	23	34

NDNS: UK National Diet and Nutrition Survey. ^1^ Added sugars (European Food Safety Authority, EFSA): Sugar—table + Sugar—other; ^2^ Free sugars (World Health Organisation, WHO): Added sugars (EFSA) + Sugar—honey + Sugar—fruit juice; ^3^ Free sugars (UK Scientific Advisory Committee on Nutrition, SACN): Free sugars (WHO) + Sugar—pureed fruit. ^4^ Discreationary sugars added to foods and drinks, apart from tea and coffee; ^5^ Other milks include milkshake, coffee whitener, milk drinks, hot chocolate, milk alternatives, dried milk and milks other than cows’ milk. ^6^ Soft drinks include squashes, cordials, tonic water, energy drinks, all types of still and carbonated soft drinks and ice tea. ^7^ Diet refers to low-calorie, no-added-sugar (EFSA) and sugar-free varieties. ^8^ Beverages dry weight includes instant and powder forms of drinking chocolate, cocoa, malted drinks, milk shake powder, tea and coffee such as cappuccino, latte and mocha with sugar. *Some food groups such as meats are removed as they don’t have relevance to this table. Therefore, the sum of% contributions may not always add up to% consumers.

**Table 7 nutrients-12-00393-t007:** Contribution (%) of NDNS RP food group sources to intake of sugars by age (19–64 Years) (NDNS 2014–2016)*.

	Adults, 19–64 Years (*n* = 1082)							
	Added Sugars (EFSA) ^1^	Free Sugars (WHO) ^2^	Free Sugars (SACN) ^3^	Sugar—Honey	Sugar—Fruit Juice	Sugar—Pureed Fruit	Sugar—Stewed Fruit	Sugar—Dried Fruit
Cereals and cereal products	31.6	25.3	25.1	11.5	2.6	10.2	14.7	32.6
Breakfast cereals	5.9	4.8	4.7	7.0	0.3	0.4	0.2	13.0
Biscuits, buns, cakes, pastries, fruit pies	20.5	16.6	16.5	3.6	1.5	9.8	14.1	18.3
Pasta, rice, bread and other cereals	5.2	3.8	3.8	0.8	0.8	0.0	0.4	1.2
Sugars, preserves, confectionery	29.3	26.0	25.7	9.1	2.0	12.4	0.1	1.3
Sugars ^4^, preserves, sweet spreads	5.6	6.1	6.1	8.7	0.7	11.2	0.0	0.0
Sugars consumed in tea and coffee	13.0	10.9	10.8	0.0	0.0	0.0	0.0	0.0
Sugar confectionery	1.9	1.7	1.7	0.1	1.2	1.2	0.0	0.5
Chocolate confectionery	8.8	7.3	7.1	0.4	0.1	0.0	0.1	0.9
Milk and milk products	7.1	5.9	6.0	0.7	1.8	15.7	3.6	0.0
Yogurt, fromage frais, dairy desserts	3.4	2.8	2.9	0.7	0.2	13.7	3.4	0.0
Ice cream	2.8	2.2	2.2	0.0	0.3	1.9	0.0	0.0
Cream and other milks ^5^	1.0	0.9	0.9	0.0	1.6	0.1	0.4	0.0
Fruits and fruit products	0.2	0.2	0.4	0.0	0.4	2.3	4.2	9.1
Non-alcoholic beverages	16.6	21.4	21.8	0.0	41.0	12.8	0.7	0.0
Fruit and vegetable juice	0.0	6.3	6.5	0.0	23.4	3.4	0.0	0.0
Soft drinks ^6^ (not diet ^7^)	16.3	14.5	14.4	0.0	10.0	1.0	0.7	0.0
Soft drinks ^6^ (diet ^7^)	0.0	0.5	0.7	0.0	7.4	8.4	0.0	0.0
Alcoholic beverages	0.7	9.6	9.5	0.0	31.5	0.0	0.0	0.0
Miscellaneous	6.9	5.6	5.5	1.6	6.2	7.3	2.1	2.9
Beverages dry weight ^8^	1.2	1.1	1.1	0.0	0.0	0.0	0.0	0.0
Soups	0.7	0.5	0.5	0.1	0.0	0.0	0.0	0.0
Savoury sauces and condiments	4.4	3.6	3.5	1.5	6.2	7.3	2.1	2.7
Bases (unweighted)	1082	1082	1082	1082	1082	1082	1082	1082
% consumers	100	100	100	27	91	63	29	48

NDNS: UK National Diet and Nutrition Survey. ^1^ Added sugars (European Food Safety Authority, EFSA): Sugar—table + Sugar—other; ^2^ Free sugars (World Health Organisation, WHO): Added sugars (EFSA) + Sugar—honey + Sugar—fruit juice; ^3^ Free sugars (UK Scientific Advisory Committee on Nutrition, SACN): Free sugars (WHO) + Sugar—pureed fruit. ^4^ Discreationary sugars added to foods and drinks, apart from tea and coffee; ^5^ Other milks include milkshake, coffee whitener, milk drinks, hot chocolate, milk alternatives, dried milk and milks other than cows’ milk. ^6^ Soft drinks include squashes, cordials, tonic water, energy drinks, all types of still and carbonated soft drinks and ice tea. ^7^ Diet refers to low-calorie, no-added-sugar and sugar-free varieties. ^8^ Beverages dry weight includes instant and powder forms of drinking chocolate, cocoa, malted drinks, milk shake powder, tea and coffee such as cappuccino, latte and mocha with sugar. *Some food groups such as meats are removed as they don’t have relevance to this table. Therefore, the sum of % contributions may not always add up to % consumers.

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
