# Peer review of "Free and Added Sugar Consumption and Adherence to Guidelines: The UK National Diet and Nutrition Survey (2014/15–2015/16)"

_nutrients, 2020, doi:10.3390/nu12020393_

Round 1
Reviewer 1 Report
Thank you for your comments and modifications.
Author Response
Dear Reviewer,
Thank you for your time to review our manuscript and supporting the publication.
Reviewer 2 Report
Free and Added Sugar Consumption and Adherence 2 to Guidelines: The UK National Diet & Nutrition 3 Survey (2014/15–2015/16)
A very interesting work of Amoutzopoulos and colleagues, focusing on total sugar intake of different food groups in different age groups between 2014 and 2016 in UK. Although a very nice approach, the following points have to be mentioned:
1. In introduction (line 34), it has to be pointed out that almost all of these metabolic disease (except caries) are mediated by excess energy intake, since sugars per se do not affect body weight gain, type 2 diabetes or other diseases, if isocalorically exchanged with other carbohydrates.
These findings are from systematic reviews and meta-analyses with the highest scientific evidence:
Body weight: Te Morenga et al. 2013, BMJ
Body weight: Fattore et al. 2017, Am J Clin Nutr
Glycemic Control: Choo et al. 2018, BMJ
CVDs: Schwingshackl et al. 2019, Am J Clin Nutr
2. In discussion, if discussing the SACN recommendations please also discuss the findings by Louie and colleagues, indicating that an intake of free sugar intake of less than 5 En% can be accompanied by micronutrient deficiency in an Australian population:
https://academic.oup.com/ajcn/article/107/1/94/4825205
https://www.ncbi.nlm.nih.gov/pubmed/30066176
Also, mortality-rate can be increased if free sugar intake is too low, as shown in a Swedish population:
https://www.ncbi.nlm.nih.gov/pubmed/30590448
Author Response
Dear Reviewer
Response: Many thanks for further suggestions and reference recommendations. Please find our responses below.
Free and Added Sugar Consumption and Adherence 2 to Guidelines: The UK National Diet & Nutrition 3 Survey (2014/15–2015/16)
A very interesting work of Amoutzopoulos and colleagues, focusing on total sugar intake of different food groups in different age groups between 2014 and 2016 in UK. Although a very nice approach, the following points have to be mentioned:
In introduction (line 34), it has to be pointed out that almost all of these metabolic disease (except caries) are mediated by excess energy intake, since sugars per se do not affect body weight gain, type 2 diabetes or other diseases, if isocalorically exchanged with other carbohydrates.
These findings are from systematic reviews and meta-analyses with the highest scientific evidence:
Body weight: Te Morenga et al. 2013, BMJ
Body weight: Fattore et al. 2017, Am J Clin Nutr
Glycemic Control: Choo et al. 2018, BMJ
CVDs: Schwingshackl et al. 2019, Am J Clin Nutr
Response: The authors thank to reviewer for the suggestion and references. We have expanded the intro to make it a bit more rounded. However we have avoided being very specific as the aim of our paper is not to discuss the health impact of sugar intake in such detail.
In discussion, if discussing the SACN recommendations please also discuss the findings by Louie and colleagues, indicating that an intake of free sugar intake of less than 5 En% can be accompanied by micronutrient deficiency in an Australian population:
https://academic.oup.com/ajcn/article/107/1/94/4825205
https://www.ncbi.nlm.nih.gov/pubmed/30066176
Also, mortality-rate can be increased if free sugar intake is too low, as shown in a Swedish population:
https://www.ncbi.nlm.nih.gov/pubmed/30590448
Response: Thank you for your time to recommend these references. The authors have the opinion that the discussion of this paper is about estimating sugar consumption in surveys using different definitions of sugar, not specifically sugars and health. Therefore authors are reluctant to add this information into paper as they didn’t think the discussion had scope for this level of detail.
This manuscript is a resubmission of an earlier submission. The following is a list of the peer review reports and author responses from that submission.
Round 1
Reviewer 1 Report
This manuscript presents the results of a study where the authors estimate the sugar intake of the UK population using a recent developed method. This method provides data for different sugars. Further, they demonstrate the difference in using different definitions of sugar (added sugar and free sugar) and discuss the impact of using one over the other on the intake estimates.
The authors have done a great job in writing the paper; all sections are very well-written, thorough and easy to read. I have a few comments, which I ask the authors to address. Thereafter, I recommend the paper ready to be published.
Specific comments
Abstract
Please state clearly that the terms “added sugar” and “free sugar” are used for different sugar definitions and includes different sugar sources.
Line 22: Please indicate the proportion of people meeting the resommendations instead of using the imprecise phrase “relatively low”.
Line 25: in relation to my first comment, make it clear that you are dealing with different definitions and depending on which definition you use the intake differs.
Introduction
Line 52: I miss the definitions of free and added sugar. I know that you present them further down, but it would be good to have them presented here.
Materials and methods
Line 89-91: This sentence is difficult to read, please rephrase.
Line 100: This method should be summarized here for the convenience of the reader
Line 111: You mention that consumers should report amount of food consumed using different measures. Please, discuss these methods in the discussion section
Results
Section 3.2: Please also indicate the sugar source contributing the least to overall sugar intake
Section 3.3: I am surprised to read that this few consumers meet the recommendations when the average intake is close to the recommendation. Please include this in your discussion of the results.
Further, please also indicate if the proportion of individuals whom meet the recommendations were significantly different between definitions.
Discussion
On page 16, you mention that the inclusion of fruit puree into the definition of free sugar has minimal impact on population level estimats. However, could this overall conclusion be biased by how much fruit puree consumers eat?
Conclusion
Like in the abstract, please provide numbers when you mention the proportion of people meeting the free sugar recommendations instead of using the phrase “relatively low”.
Reviewer 2 Report
Please give some indication, probably in the introduction, that the term "sugars" as applied to human diets is a collective term for several different chemical species. Thus "table sugar" is essentially pure sucrose whereas fruit juice, honey and syrups contain mixtures of sucrose, glucose and fructose, and often oligosaccharides of different size as well. These compounds are invariably lumped together as "sugars" so the point doesn't need to be laboured, but it should be acknowledged. p. 3, line 110 - "...data were collected." p. 4, lines 140 to 141. Surely less than/greater than symbols are the wrong way round, and should read EFSA < WHO < SACN p. 4, lines 150 -152. I do not see the need to attach "+" signs to these numerical differences, unless the direction of the comparison is specified. p. 4, lines 144, 149 - 150 Should read "The difference... was (or was not) statistically significant (p<0.05)...." p. 10 I find much of paragraph 3.3. confusing. Surely the proportion in each population group achieving the WHO recommendations should always be higher than the proportion achieving the SACN recommendation, because the latter is more stringent? This is not what the text states. Also, in the brackets stating those percentages e.g. (15% and 54-53%) it should be made clear that the 54-53% percentages apply to males and females respectively, if indeed that is what is meant. p. 16 What is meant by "for their concentration of sugars coming from natural sources." What exactly is the definition of natural sources of sugar used in this context? p.16 para 4. The term "unhealthy foods" should be used with caution in a paper of this nature. In paragraph 4 ("underreporting unhealthy foods") should, in my opinion, read "underreporting foods perceived as unhealthy.)